# Low-Dose Administration of Cannabigerol Attenuates Inflammation and Fibrosis Associated with Methionine/Choline Deficient Diet-Induced NASH Model via Modulation of Cannabinoid Receptor

**DOI:** 10.3390/nu15010178

**Published:** 2022-12-30

**Authors:** Nouf Aljobaily, Kelsey Krutsinger, Michael J. Viereckl, Raznin Joly, Bridger Menlove, Brexton Cone, Ailaina Suppes, Yuyan Han

**Affiliations:** 1Division of Microbiology and Immunology, Department of Pathology, University of Utah, Salt Lake City, UT 84112, USA; 2Department of Biological Sciences, University of Northern Colorado, Greeley, CO 80639, USA

**Keywords:** non-alcoholic steatohepatitis, non-alcoholic fatty liver disease, cannabigerol, methionine-choline-deficient diet, inflammation, fibrosis

## Abstract

Non-Alcoholic Steatohepatitis (NASH) is the progressive form of Non-Alcoholic Fatty Liver Disease (NAFLD). NASH is distinguished by severe hepatic fibrosis and inflammation. The plant-derived, non-psychotropic compound cannabigerol (CBG) has potential anti-inflammatory effects similar to other cannabinoids. However, the impact of CBG on NASH pathology is still unknown. This study demonstrated the therapeutic potential of CBG in reducing hepatic steatosis, fibrosis, and inflammation. Methods: 8-week-old C57BL/6 male mice were fed with methionine/choline deficient (MCD) diet or control (CTR) diets for five weeks. At the beginning of week 4, mice were divided into three sub-groups and injected with either a vehicle, a low or high dose of CBG for two weeks. Overall health of the mice, Hepatic steatosis, fibrosis, and inflammation were evaluated. Results: Increased liver-to-body weight ratio was observed in mice fed with MCD diet, while a low dose of CBG treatment rescued the liver-to-body weight ratio. Hepatic ballooning and leukocyte infiltration were decreased in MCD mice with a low dose of CBG treatment, whereas the CBG treatment did not change the hepatic steatosis. The high dose CBG administration increased inflammation and fibrosis. Similarly, the expression of cannabinoid receptor (CB)1 and CB2 showed decreased expression with the low CBG dose but not with the high CBG dose intervention in the MCD group and were co-localized with mast cells. Additionally, the decreased mast cells were accompanied by decreased expression of transforming growth factor (TGF)-β1. Conclusions: Collectively, the low dose of CBG alleviated hepatic fibrosis and inflammation in MCD-induced NASH, however, the high dose of CBG treatment showed enhanced liver damage when compared to MCD only group. These results will provide pre-clinical data to guide future intervention studies in humans addressing the potential uses of CBG for inflammatory liver pathologies, as well as open the door for further investigation into systemic inflammatory pathologies.

## 1. Introduction

Non-Alcoholic Steatohepatitis (NASH), denoted by hepatic fibrosis and inflammation, is the progressive form of Non-Alcoholic Fatty Liver Disease (NAFLD) [1,2]. Globally, NAFLD is one of the most common liver diseases, affecting nearly a quarter of the world’s population [3,4,5]. The progression from NAFLD to NASH begins with excessive fat deposition in hepatocytes, causing steatosis and eventually inflammation to the point of hepatitis [1,6]. These changes are reversible, but when inflammation becomes chronic, fibrosis starts to develop and progress, which can lead to permanent liver damage [7,8]. Currently, there are no Food and Drug Administration (FDA)-approved treatments for NAFLD, and patients are limited to lifestyle modifications, including restrictive diets and managing other risk factors or comorbidities such as type 2 diabetes and cardiovascular disease-related complications [9,10]. Even more, liver transplant, with all the associated complications, is still the most common way to treat advanced NASH [11]. Thus, it is critical to increase our efforts in finding possible treatments to slow the progression of NASH. While many clinical trials are in the works to test the effects of known metabolic disease medications and supplements, such as metformin and vitamin E, the lack of widespread success in these studies suggests the need for further investigations into novel substances that might provide insightful therapeutic potential [10,12,13,14,15,16,17].

Cannabis and cannabinoids have resurfaced in research for their anti-inflammatory effects, as a treatment for epilepsy, and even in their roles as receptor agonists [18,19,20,21,22]. In the last few years, studies have shown Cannabigerol (CBG) and other cannabinoids to be potentially successful in the treatment of neurodegenerative diseases, to have protective properties against oxidative stress induced damage in intestinal epithelial cells, and even success in the treatment of cancer cells [18,19,22]. However, CBG has not been investigated before in liver disease [23]. Therefore, this study aims to evaluate the efficacy of CBG in alleviating NASH pathology.

Previously, endocannabinoid receptors were shown to act as the main biological mediators of cannabis compounds [24]. In the 1990′s, cannabinoid (CB) receptors were first identified in the human body [19], which was followed by subsequent investigations suggesting a dependent relationship between CB receptors and the development of metabolic diseases [25]. In the liver, CB receptors 1 and 2 have been shown to play a role in liver pathologies such as NAFLD, NASH, and hepatocellular carcinoma [25,26]. Specifically, activation of the CB1 receptor results in increased lipid storage and fibrosis, but activation of the CB2 receptor results in steatosis and more mild hepatocyte injury [27,28].

Using the methionine/choline-deficient (MCD) diet (a well cited NASH animal diet model) [29,30,31] and its corresponding control (Table 1), this study aims to investigate the impact of CBG supplementation on the main pathologic features of NASH: hepatic steatosis, inflammation, and fibrosis [3].

## 2. Materials and Methods

### 2.1. Diet, Reagents, and Primers

Control (CTR) (TD.94149) and MCD diet (TD.90262) were purchased from Envigo (Denver, CO, USA), while all reagents and primers were synthesized by Thermo Fisher Scientific (Denver, CO, USA), VWR (Radnor, PA, USA), Biolegend (San Diego, CA, USA), and Invitrogen (Fredrick, MD, USA) unless otherwise indicated. CBG powder was generously provided by Mile High Labs (Broomfield, CO, USA).

### 2.2. Animals, Diets and Cannabigerol Treatments

All proposed procedures have been approved by the Institutional Animal Care and Use Committee at the University of Northern Colorado (protocol no. 1910CE-YH-M-22). Eight-week-old male C57BL/6 mice (*n* = 36) were used in this study. Mice were randomly assigned into either the control group (*n* = 18) or the MCD diet group (*n* = 18) and were fed the CTR diet or MCD diet for 3 weeks. Then, they were randomized into three-subgroups and intraperitoneally (I.P.) injected with a vehicle (2.5% dimethyl sulfoxide in saline), a low dose (2.46 mg/kg/day, L. CBG) or a high dose (24.6 mg/kg/day, H. CBG) of CBG three times a week for two additional weeks. The high dose concentration was based on one-tenth of the human maximum allowed daily consumption of Cannabidiol (CBD) and then converted into mice based on previously published study [32]. The low-dose concentration was chosen based on previously published study [33]. During the length of the trial, body weight and food consumption were measured 3 times a week. All animals were euthanized on the last day of the study.

To prepare the H. CBG treatment, 6.642 mg of anhydrous CBG was dissolved in Dimethyl sulfoxide (DMSO) and then in 1.5 mL phosphate-buffered saline (PBS). The Low CBG treatment was tenth the H. CBG. The control treatment was prepared in the same way as the high dose CBG without the CBG.

### 2.3. Liver Harvest

Upon sacrifice, liver tissues were harvested and weighed. Then, they were either embedded in Tissue-Tek O.C.T Compound and snap frozen in liquid nitrogen or directly snap frozen in liquid nitrogen. All samples were stored at −80 °C until used, except tissues used for RNA-related analyses, which were stored at 4 °C.

### 2.4. Histological Staining

To evaluate overall liver health, hematoxylin and eosin (H&E) staining was performed. 8 µm frozen liver sections were fixed in 10% neutral buffered formalin (NBF) for 10 min, followed by a wash in 95% ethanol and 2–3 changes of tap water rinsing. Tissues were then stained with hematoxylin for 40 s and washed in ammonia water. They were then mordant in 95% ethanol followed by eosin stain for 10 s. After that, they were dehydrated in 2 changes of 95% ethanol followed by 3 changes of 100% ethanol. Finally, tissues were washed in 2 changes of xylene and mounted with a mounting medium. All images were taken at 20× magnification.

Lipid accumulation was evaluated using Oil Red O staining. 8 µm frozen liver sections were fixed in 10% NBF for 10 min then immediately washed in 3 changes of distilled water. Tissues were then placed in 100% propylene glycol for 5 min followed by pre-heated Oil Red O staining for 8 min. After that, they were placed in 85% propylene glycol for 5 min and rinsed in 2 changes of distilled water. Tissues were then counterstained with hematoxylin for 40 s and washed afterward in running tap water for 3 min before finally being mounted in aqueous mounting media. All images were taken at 20× magnification.

Liver fibrosis was assessed using Picro-Sirius Red staining. 8 µm frozen liver sections were fixed in 10% NBF for 10 min followed by xylene for 10 min. Then, they were rehydrated in 100%, 90%, and 70% ethanol. The nucleus was stained with hematoxylin for 40 s followed by rinsing in tap water for 10 min. After, tissues were stained for collagen deposition using Picro-Sirius Red for 1 h followed by 2 washes in 0.5% acidified water. Tissues were then dehydrated in 70%, 90%, and 100% ethanol before being placed in xylene and then fixed in mounting media.

### 2.5. Gene Expression

Total RNA was extracted using Pure Link™ RNA Mini Kit obtained from Invitrogen (Waltham, MA, USA) and converted into cDNA using High-Capacity cDNA Reverse Transcription Kit obtained from Applied Biosystems™ (Waltham, MA, USA) according to manufacturer recommendations. RT-qPCR was performed to analyze the mRNA expression of CD36 (cluster of differentiation 36, fatty acid transferase), α-SMA (alpha smooth muscle actin), TGF-β1 (transforming growth factor beta 1) (Table 2). Six biological replicates were used to measure relative gene expression at the transcriptional level unless otherwise indicated. All gene expressions were normalized to glyceraldehyde-3-phosphate dehydrogenase (GAPDH).

### 2.6. Immunofluorescence Staining

Bodipy or other immunofluorescent staining was used to evaluate the lipid accumulation or gene expression, respectively. 8 µm frozen liver sections were fixed in 10% NBF for 10 min then washed with PBS. Sections were then blocked for 20 min with 10% normal goat serum diluted in PBS, followed by a rinse in PBS. Thereafter, sections were incubated with bodipy dye for 30 min or with primary antibody overnight in a 4 °C incubator. The next day, specimens were washed with PBS and stained with the secondary antibody for 45 min in the dark at room temperature. Finally, tissues were mounted with DAPI mounting medium and stored in −20 °C until analyzed with a Zeiss 700 confocal microscope. All images were taken at 20× magnification and quantified using Fiji ImageJ software. Detailed antibody information is listed in Table 3.

### 2.7. Triglyceride Assay

Triglyceride levels were measured using commercially available kit from Cayman Chemical (Ann Arbor, MI, USA). The assay was performed according to manufacturer instructions. Briefly, 350–400 mg of snap frozen liver tissue was washed with cold PBS and homogenized in 2 mL of the Nonidet P-40 (NP40) substituted assay reagent using a homogenizer. The mixture was then centrifuged at 10,000× *g* for 10 min at 4 °C to remove insoluble material. The supernatant was collected and diluted two-fold using the diluted NP40 substitute assay reagent. 10 µL of each of the standard solutions (available in the kit) and samples were added in respective wells of the assay-plate along with 150 µL of the diluted enzyme mixture solution (available in the kit). The plate was incubated for 30 min at 37 °C and the absorbance was measured at 550 nm using a plate reader.

### 2.8. Statistical Analysis

All data were analyzed using GraphPad Prism 9 software (San Diego, CA, USA) and reported as mean ± SEM (Standard Error of the Mean). One-way analysis of variance (ANOVA) was performed to test the significant difference between group means for each experiment in this study. After performing ANOVA tests, Tukey’s post hoc tests were performed comparing every pair of groups to indicate which group was causing the significance in the ANOVA tests, when significance was detected. For all tests used, a *p*-value ≤ 0.05 was considered significant.

## 3. Results:

### 3.1. Low Dose of CBG Treatment Alleviated MCD Diet-Induced NASH Symptoms in C57BL/6 Mice

We first characterized the overall liver health to evaluate the effect of CBG in the MCD diet-induced NASH mouse model. As indicated in Figure 1A, livers appeared pale in color and smaller in size in mice fed with MCD diet without CBG treatment. Conversely, liver morphology was much improved after the low dose of CBG treatment in MCD group. We did not observe significant change in food consumption among all groups, although there was a trend toward less food consumption in MCD group when compared to other groups (Figure 1B). Furthermore, the overall body weight in all MCD groups was significantly lower when compared with mice fed with CTR diet (Figure 1C), while the liver weight of MCD was not significantly reduced when compared to CTR (Appendix A). The liver weight to body weight ratio was significantly increased in MCD group without CBG treatment and was reversed to normal with low dose of CBG treatment (Figure 1D). Interestingly, high dose of CBG treatment failed to show improvement of liver health from the MCD only treatment. To further understand whether the overall liver health was impacted by either the MCD diet or the CBG treatment, liver tissues were subjected to H&E staining. While no significant damage was shown in CTR diet groups, hepatic ballooning and inflammation loci were shown in MCD only mice. Hepatic ballooning and leukocyte infiltration were alleviated in MCD L.CBG group while the alleviation was not as significant in MCD H.CBG group (Figure 1E). Overall, our results suggest that the low dose of CBG treatment has a higher potential for alleviating liver damage due to MCD-induced NASH symptoms, compared to high dose of CBG which caused elevated liver damage.

### 3.2. CBG Treatment Did Not Change the Steatosis Caused by MCD Diet

We next investigated whether CBG would alleviate the MCD-induced steatosis and whether different doses have different impacts on steatosis. To determine the effect of CBG on steatosis we stained frozen liver sections with Bodipy dye. We found an increase of lipid accumulation in all MCD-fed groups, though CBG was not able to lower the lipid amount in either the MCD L. CBG or MCD H. CBG group (Figure 2A,B). Oil Red O staining was performed to further explore the lipid accumulation (Appendix A). Comparable results were observed: where there was accumulation of lipids in the MCD-fed groups, but no or minimal lipid accumulation was observed in CTR-fed groups.

CD36 is a widely used biomarker for NASH as it is involved in the transfer of lipids into cells [34,35]. We measured the gene expression of CD36 using RT-qPCR (Figure 2C). There was an increase in gene expression of CD36 in the liver in the MCD diet groups when compared to CTR diet group, which is consistent with the Bodipy staining. However, there was no significant difference found among different MCD diet groups. Elevated triglyceride levels have been widely used to aid in the screening of non-symptomatic NAFLD and NASH patients in peripheral blood [36]. Therefore, we measured the triglyceride levels using a commercially available ELISA (enzyme-linked immunoassay) kit (Figure 2D). As expected, triglyceride levels increased in the liver tissues of the MCD diet groups when compared to CTR group. Consistent with the CD36 expression, there was no significant difference noted between CBG subgroups within their respective diets. Considering this, neither the low dose of CBG nor the high dose of CBG alleviated the steatosis profile in the liver.

### 3.3. Hepatic Inflammation Was Reduced with the Low Dose of CBG, While the High Dose of CBG Enhanced Inflammation in Liver Tissues

Enhanced hepatic inflammation is one of the NASH features, therefore a potential target to treat NASH is to reduce the inflammation. CBG has shown an anti-inflammatory effect in other diseases. Therefore, we evaluated the potential anti-inflammatory effect of CBG in the MCD-induced NASH mouse model. Initially, immunofluorescence staining for CD45 (Cluster of differentiation 45), a marker for leukocytes (white blood cells), was utilized to measure hepatic inflammation (Figure 3A,B). As expected, the infiltration of leukocytes (CD45^+^) was increased in mice fed with MCD diet, while the low dose of CBG significantly diminished the infiltration of leukocytes in mice fed with MCD diet. However, the high dose of CBG failed to lower the leukocyte infiltration in mice fed with MCD diet. In fact, the proliferation of immune cells in the group treated with high dose of CBG showed similar levels as the groups treated with MCD diet only.

Since we observed that CBG could inhibit white blood cell infiltration in MCD-fed mice, we next investigated which type of immune cells is the primary contributor to this change. Studies have shown that macrophages are an important regulator in the pathogenesis of NASH [37]. Inhibition of macrophage infiltration leads to alleviation of NASH-related fibrosis [38]. Similar to the CD45^+^ staining in Figure 3A,B, the low dose of CBG reduced the macrophage population (F4/80^+^, a macrophage marker) triggered by MCD diet. Meanwhile, the high dose of CBG did not show a reduction in macrophage population in MCD-fed mice (Figure 3C,D). There was a significant increase in F4/80 mRNA levels in MCD H. CBG group when compared to both the CTR and MCD groups. However, no change was observed in mRNA level of F4/80 with low dose of CBG treatment in MCD mice (Figure 3E). Combined, this indicates the F4/80 expression was enhanced by high dose of CBG in MCD mice liver. Our data suggested that low dose of CBG can ameliorate hepatic inflammation, while high dose of CBG was not protective against MCD-induced inflammation.

### 3.4. A Low Dose of CBG Ameliorated Hepatic Fibrosis; While High Dose of CBG Increased Hepatic Fibrosis

Another main pathological feature of NASH is liver fibrosis, as a result of chronic inflammation [6] Chronic steatosis can induce inflammation and permanent fibrosis leading to restricted liver function [2]. Hence, hepatic fibrosis was evaluated using Picro-Sirius Red staining, which stains for collagen in the liver tissues; the redness intensity was evaluated using ImageJ software. There was a significant reduction in collagen deposition in the MCD L. CBG group when compared to the MCD only group (Figure 4A,B). There was a significant increase in the collagen deposition in the CTR H. CBG group and MCD H. CBG when compared to the CTR group. These results were confirmed using frozen liver sections stained with alpha smooth muscle actin (α-SMA) (Figure 4C,D). Similarly, α-SMA mRNA gene expression showed the low dose of CBG treatment decreased expression of α-SMA when compared to MCD (Figure 4E). Taken together, consistent with our previous finding in Figure 3, hepatic fibrosis was alleviated with a low dose of CBG but not with the high dose of CBG.

### 3.5. CB Receptors Showed Similar Trends Asthe Inflammatory Response in MCD-Fed Mice, and the Low Dose of CBG Alleviated the Expression of CB Receptors

Both Cannabinoid (CB) 1 and 2 receptors were previously shown to interact with CBG [39,40]. CB receptors 1 and 2 were shown to play a role in liver diseases such as NAFLD, NASH, and hepatocellular carcinoma [25,26]. Thus, we sought to investigate whether CBG alleviated MCD-induced NASH occurs via regulation of CB receptors. First, we assessed the expression of CB receptors in frozen liver sections (Figure 5A,B,D). Both CB receptors showed increased expression in the MCD groups when compared to the CTR group, while there was a decrease in both CB1 and CB2 mRNA expression in MCD L. CBG group.

Additionally, because activation of CB receptors inhibits the degranulation of mast cells, we assessed the co-expression of FcεR1, the mast cell biomarker, binding along with CB receptors [41]. We found co-localization of both CB1 and CB2 with FcεR1 within all MCD groups (Figure 5C). The expression levels of CB1 and CB2 are consistent with Figure 5A, where the FcεR1^+^ cells were increased in MCD and MCD H. CBG groups. The FcεR1^+^ cell numbers decreased upon the treatment of the low dose of CBG in MCD group. The CB1 receptor did not completely co-localize with FcεR1^+^ cells in MCD H. CBG group indicating the high dose of CBG in MCD might trigger other cell types to express the CB1 receptors. Taken together, the low-dose treatment of CBG inhibits the expression of CB1 and CB2 receptors in mast cells that was stimulated by MCD diet. This inhibitory effect was not seen in the high dose of CBG treatment.

### 3.6. The Low Dose of CBG, Not the High Dose of CBG, Reduced Mast Cell Activation in MCD Diet via TGF-β1 Signaling

Since we found mast cell numbers decreased upon the treatment of low-dose CBG in MCD, we further evaluated which proinflammatory factor might mediate the activation of mast cells. Based on previous findings, TGF-β1 was shown to activate mast cells and cholangiocytes to induce fibrosis [42]. Therefore, we co-stained TGF-β1 with a mast cell biomarker (FcεR1) and a cholangiocytes biomarker (CK19, Cytokeratin 19), respectively. Enhanced ductal reaction and activated mast cells were shown in MCD mice (Figure 6A,B).

The expression of TGF-β1 was predominantly co-localized with mast cells in MCD diet (Figure 6B), but not co-localized with cholangiocytes (Figure 6A). The low CBG administrated decreased the expression of TGF-β1 as well as the mast cell numbers in MCD diet mice. Further, the co-localization of mast cells and TGF-β1 was significantly enhanced in MCD-diet mice with the high dose of CBG treatment (Figure 6B). Although the cholangiocyte proliferation was enhanced in H. CBG with MCD treatment, TGF-β1 was not co-localized with CK19^+^ cells. This suggests that the cholangiocytes proliferation, induced by high dose CBG in MCD diet, is TGF-β1 independent. Taken together, the mast cells activation is related to TGF-β1 pathway and plays a more significant role in mediating the regression of MCD upon the CBG treatment when compared to cholangiocytes.

## 4. Discussion

In the past few decades, metabolic diseases have become one of the most prevalent global health concerns [3,11]. NAFLD, being the hepatic manifestation of metabolic syndrome, has surpassed all other liver disorders, with NASH amounting to nearly a fourth of these cases [4]. While lifestyle changes and invasive surgeries are the only approved clinical approaches to slow the progression of NASH, it is critical to investigate alternative treatments and strategies [2,9,10,11,17,43]. Permanent fibrotic damage to the liver can lead to cirrhosis and even hepatocellular carcinoma if progression is not halted [6,7,8]. The pathologic features of NASH are interconnected as hepatic steatosis leads to chronic inflammation, which eventually results in fibrosis [44,45]. It is important to acknowledge that the MCD diet does not fully replicate the human NASH pathology. The MCD diet induced NASH mouse model fails to show metabolic syndrome, glucose intolerance, insulin resistance and body weight gain, though it replicates other NASH liver injury such as hepatic steatosis, inflammation, ER stress, cellular apoptosis and fibrosis over a short period [7,46,47]. Itis important to evaluate the systemic disease markers, such as bilirubin, aspartate aminotransferase and Alanine aminotransferase, for the therapeutic potential of CBG to improve liver function. In this study, we only analyzed the male mice, thus the therapeutic effect of CBG in females needs further investigation. In the exploratory experiments, we have found gender differentiated responses to the CBG in terms of alleviation of inflammation and fibrosis. However, further experiments need to be done to confirm the results. Taken together, we have demonstrated the therapeutic potential of CBG in NASH mouse model in male mice, as it reduced NASH-, related pathologic features when administered in a low dose.

Hepatic inflammation is a primary pathology that can lead to liver fibrosis in NASH patients [2,48]. A previous study has reported that hepatic inflammation plays an essential role in the progression of hepatic steatosis and fibrosis [6]. The findings in our study are consistent with a previous study by Aqawi et al. showing an anti-inflammatory effect of CBG [49]. Similarly, other reports found that F4/80 expression (a murine macrophage biomarker) was reduced with the intervention of atypical CBD [22]. This would support our findings that indicate the high dose of CBG might trigger increased macrophage proliferation, but not elevated F4/80 mRNA levels in each cell. Overall, the findings suggest that a low dose of CBG has a potential in reducing inflammation associated with NASH via attenuating macrophage proliferation and infiltration.

A study investigated the beneficial effect of a cannabidiol derivative using a low dose (0.05 mg/kg) of atypical CBD, showing the potential of cannabis in reducing liver fibrosis, inflammation, and oxidative stress [22]. Even though the low dose of CBG reduced fibrosis in the liver tissues, it is still important to acknowledge that the high dose of CBG had the opposite effect. This dose-dependent reaction is critical to take into consideration when administering CBG in future studies. Our study showed that CBG does not reduce hepatic steatosis, which was different from the effect of CBD in NASH [23]. Reduced circulating lipids was shown in high fat diet-CBD fed mice [23]. Conversely, clinical trials have shown no decrease in lipid accumulation in NAFLD patients with supplemental CBD, which is consistent with findings in this study [50].

We further explored the molecular mechanisms by which CBG affects the progression of MCD-induced NASH in mice. The CB1 is expressed throughout the body in peripheral tissues, with increased expression in brain [51]. In the brain, activation of CB1 promote obesity by increasing fatty acid synthesis [52]; while inhibition of CB1 showed anti-obese effect [53]. On the other hand, the CB2 receptor is primarily expressed in the immune system [19]. This study has found that the expression of CB1 and CB2 receptors showed a similar pattern to the inflammatory response, which led us to investigate which cell type is expressing these receptors. Our findings showed increased infiltration of mast cells into the liver accompanied by enhanced CB receptors activation in the MCD diet group, while low-dose CBG treatment reduced this trend. Interestingly, we noted similar trends between Figure 5A,C, suggesting that inflammatory pathways might mediate CB receptor expression. Nevertheless, further investigations are still needed to confirm the observed results. Overall, CB receptors were found to be alleviated by the low dose of CBG but not by the high dose of CBG.

Consistent with our findings, TGF-β1 signaling plays a significant role in fibrogenesis in liver diseases and is linked to the proliferation of mast cells [54]. This study also found that inhibition of mast cell TGF-β1 secretion reduced the hepatic fibrosis and cholangiocyte proliferation [54]. The CBG treatment induced the mast cell infiltration in a dose-dependent manner in MCD mice. Combine together, we could hypothesis that the low dosage CBG treatment could inhibit TGF-β1 secretion from mast cells leading to reduced hepatic fibrosis. However, further study is needed to investigate whether direct inhibition of TGF-β1 would inhibit the MCD-induced NASH symptoms or modulate the mast cells numbers with high doses of CBG treatment.

In summary, this study observed the protective effect of low-dose administration of CBG in MCD diet-induced NASH in mice. A low dose of CBG reduced hepatic fibrosis and inflammation but not hepatic steatosis, while a high dose of CBG worsened the pathologic features of NASH. In conclusion, this study provides initial findings and a foundation for future studies on the efficacy of CBG on NASH.

## Figures and Tables

**Figure 1 nutrients-15-00178-f001:**
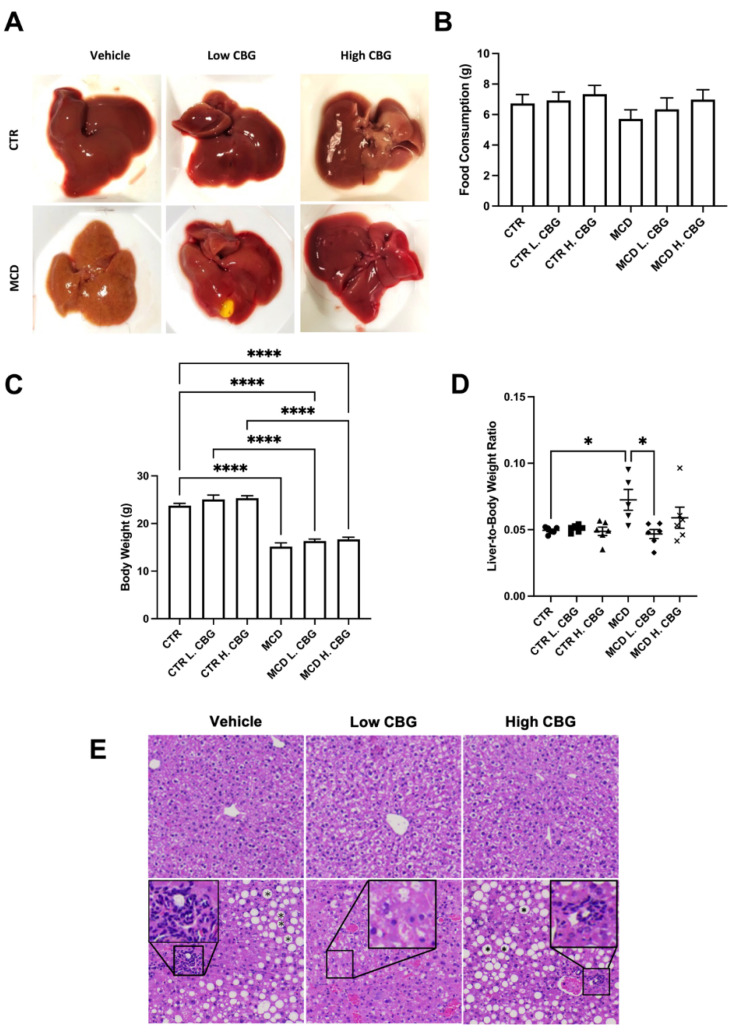
Evaluation of the overall liver health with the induction of NASH using MCD diet and role of CBG in reducing the liver damage. Representative images of liver post-harvest to illustrate the differences in morphologies between groups (**A**) Average total food consumption (**B**), Body Weight (**C**) and average liver weight-to-body weight ratio (**D**) was calculated among groups. Representation of the liver after being stained with H&E (**E**), showing lipid accumulation (mark as asterisk in hepatic ballooning area) and white blood cells infiltration (enlarged area) with the MCD diet (bottom row). * *p* < 0.05, **** *p* < 0.0001.

**Figure 2 nutrients-15-00178-f002:**
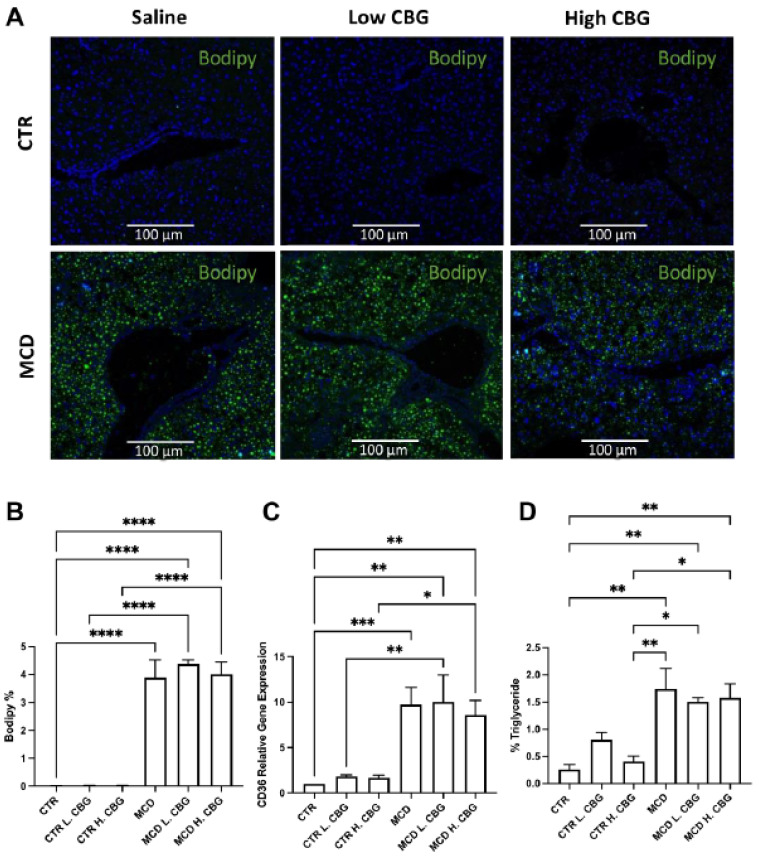
Steatosis was evaluated using histology and RT-qPCR. Immunofluorescence staining of bodipy (green) and DAPI (blue) (**A**, 20×). Immunofluorescence staining was quantified using ImageJ (**B**). mRNA expression of CD36 (**C**) was measured via RT-qPCR, and the percentage of triglyceride was evaluated using a commercially available ELISA kit (**D**). * *p* < 0.05, ** *p* < 0.01, *** *p* < 0.001, **** *p* < 0.0001.

**Figure 3 nutrients-15-00178-f003:**
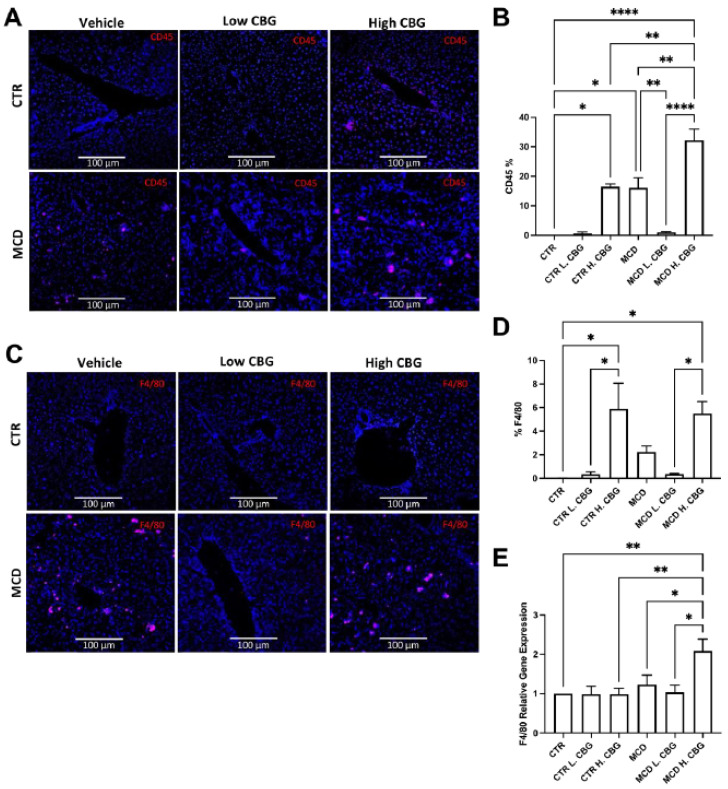
Immunofluorescence staining and RT-qPCR were utilized to measure inflammation in liver tissues. Representative image of liver frozen sections stained for white blood cells showing CD45 in red and DAPI in blue (**A**, 20×), which was quantified using ImageJ (**B**). The infiltration of macrophages (F4/80^+^) were shown in red, while nuclei were shown in blue by DAPI staining (**C**). Images of F4/80 was also quantified using ImageJ (**D**). Quantification of mRNA expression of F4/80 (**E**). * *p* < 0.05, ** *p* < 0.01, **** *p* < 0.0001.

**Figure 4 nutrients-15-00178-f004:**
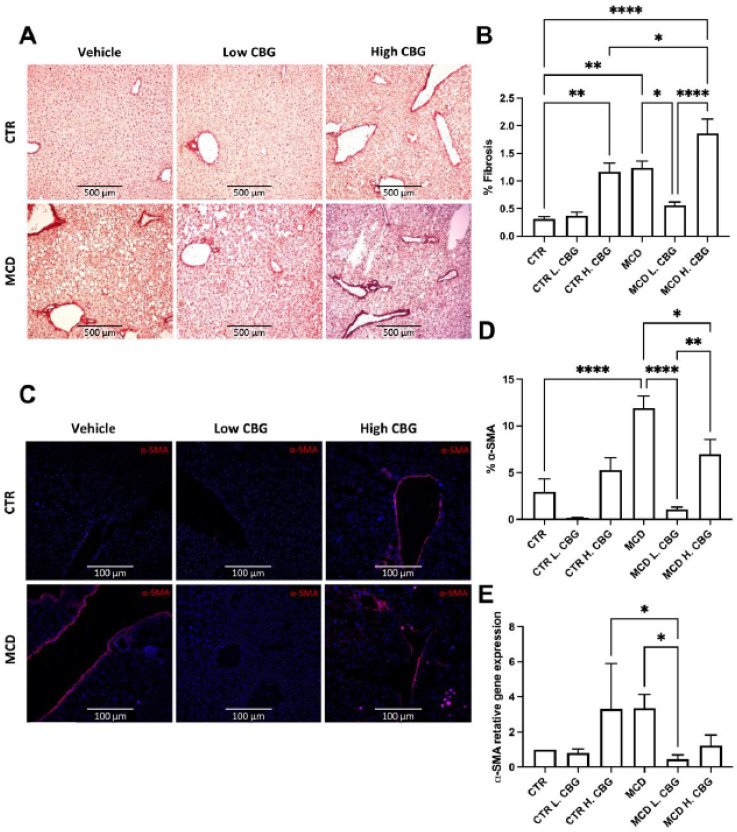
Liver fibrosis was measured by Picro-Sirius Red staining, α-SMA staining and mRNA gene expression. Representative pictures and quantification of collagen deposition in liver frozen section (**A**,**B**, 10×). α-SMA immunofluorescence staining and quantification (**C**,**D**, 20×) showing α-SMA in red and DAPI (nucleus) in blue (**D**). Quantification of α-SMA mRNA gene expression (**E**). * *p* < 0.05, ** *p* < 0.01, **** *p* < 0.0001.

**Figure 5 nutrients-15-00178-f005:**
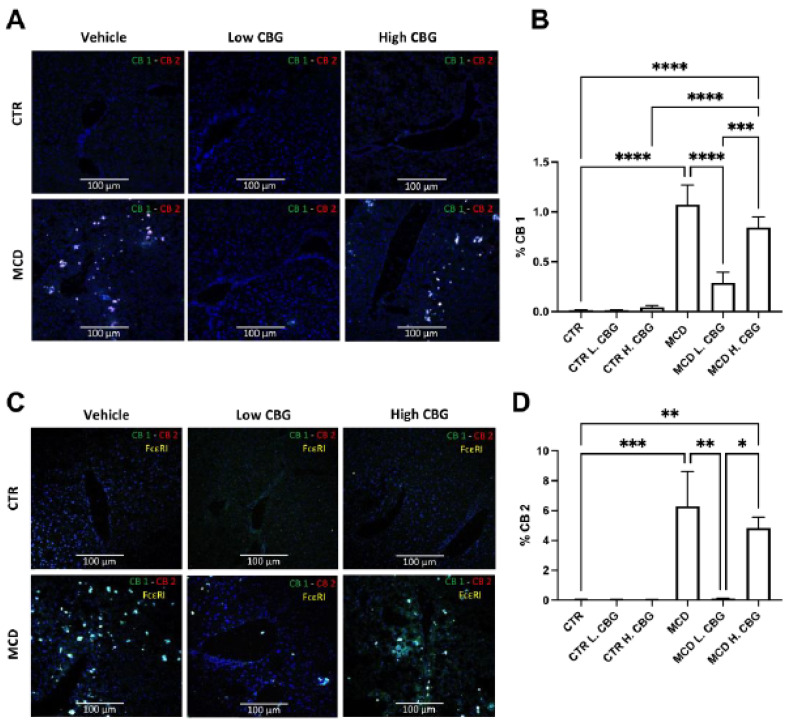
Evaluation of CB receptors with FcεR1 co-localization in liver frozen sections. Immunofluorescence staining of frozen liver sections for CB1 andCB2 (**A**, 20×), illustrating CB1 in green and CB2 in red. Then, the expression of FcεRI was overlayed to assess co-localization (**C**). % CB1 (**B**) and CB2 (**D**) were quantifies using imageJ software. * *p* < 0.05, ** *p* < 0.01, *** *p* < 0.001, **** *p* < 0.0001.

**Figure 6 nutrients-15-00178-f006:**
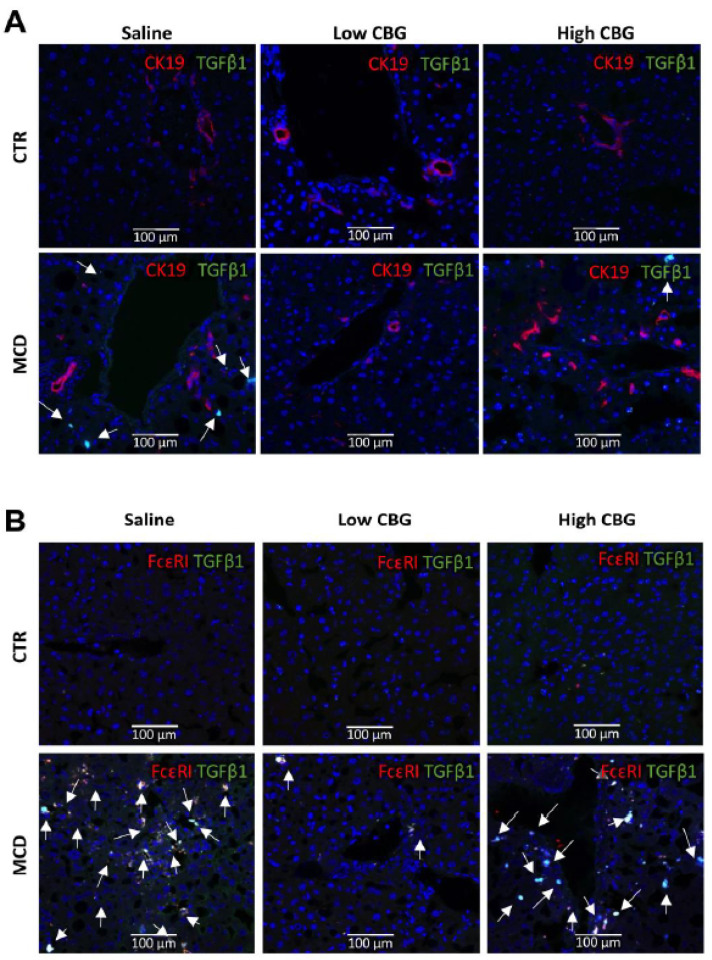
Colocalization of the proinflammatory marker TGF-β1 with the cholangiocytes (CK19^+^) or mast cells (FcεR1^+^) in mice liver tissue. (**A**) Immunofluorescence co-staining of cholangiocytes biomarker CK19 (red) and proinflammatory factor TGF-β1 (green), and DAPI (nuclei stained in blue). CK19^+^ cells were not co-localized with TGF-β1 (white arrows). (**B**) immunofluorescence co-staining of mast cell biomarker FcεR1 (red) and proinflammatory factor TGF-β1 (green), and nuclei (blue). Strong colocalization of TGF-β1 with mast cells in MCD groups (white arrows). Pictures are taken at 20× magnification.

**Table 1 nutrients-15-00178-t001:** MCD diet and MCD control diet components.

Formula	CTR	MCD
L-Amino acids (g/kg)	156.4	156.4
L-Methionine (g/kg)	8.2	0.0
Choline chloride (g/kg)	350 g/kg *	0.0
Sucrose (g/kg)	443.597	455.294
Corn starch (g/kg)	198.783	200.0
Cellulose (g/kg)	30.0	30.0
Corn oil (g/kg)	100.0	100.0
Salt mix (g/kg)	35.5	35.0
Vitamin mix (g/kg)	10.0	5.0

* Information was taken from Wagner et al.

**Table 2 nutrients-15-00178-t002:** Primer sequences for RT-qPCR.

Gene	Forward	Reverse
CD36	AATTAGTAGAACCGGGCCAC	CCAACTCCCAGGTACAATCA
α-SMA	ACTGGGACGACATGGAAAAG	AGAGGCATAGAGGGACAGCA
TGF-β1	GAGCCCGAAGCGGACTACTA	CACTGCTTCCCGAATGTCTGA
F4/80	TGACAACCAGACGGCTTGTG	GCAGGCGAGGAAAAGATAGTGT
GAPDH	TGCACCACCAACTGCTTAGC	GGCATGGACTGTGGTCATGAG

**Table 3 nutrients-15-00178-t003:** Antibodies and fluorescent dyes dilution and clones.

Antibody/ Fluorescent Dye	Company	Dilution	Clone	Catalog Number
α-SMA	Invitrogen	1:100	1A4	14-9760-83
CNR1 (CB1)	Invitrogen	1:1000	PA1-743	PA585080
CNR2 (CB2)	Invitrogen	1:200	PA5-18428	PA518428
F4/80	Biolegend	1:100	BM8	123122
Alexa Fluor 594	Biolegend	1:100	Poly4053	405326
FcεR1	Biolegend	1:200	MAR-1	134316
CD45	Biolegend	1:100	OX-1	202201
TGF-β1	Invitrogen	1:100	N/A	21898-1-AP
CK19	DSHB	1;100	N/A	TROMA-III
Bodipy	ThermoFisher	1:10,000	N/A	D3922
Alexa Fluor^®^ 647	ABCAM	1:200	N/A	ab150155
Alexa Fluor^®^ 555	ABCAM	1:200	N/A	ab150130
Alexa Fluor^®^ 488	ABCAM	1:200	N/A	ab150073

## Data Availability

Raw data and analyzed data is available upon request.

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
