# Peer review of "Low-Dose Administration of Cannabigerol Attenuates Inflammation and Fibrosis Associated with Methionine/Choline Deficient Diet-Induced NASH Model via Modulation of Cannabinoid Receptor"

_nutrients, 2022, doi:10.3390/nu15010178_

Round 1

Reviewer 1 Report

Aljobaily et al use a Methionine-Choline deficiency (MCD) dietary model in mice to induce NASH, and, using this mode study the possible beneficial effects of the non-psychotropic compound cannabigerol (CBG).

The manuscript proposes that a low dose of cannabigerol has positive effects on numerous markers of NAFLD, while not, however, ameliorating of hepatic steatosis.

Overall, this well performed investigation convincingly shows the potential benefit of CBG on the normalization of liver NASH parameters (fibrosis and inflammation), while not having a really demonstrable effect on steatosis. Also, it is noted that higher CBG doses are detrimental.

Some point will need attention from the authors:

- In all the figures presenting asterisks to denote p-values, these are actually missing.

- Rendering of figure 1 is quite poor. Please make sure to submit a source file of the same quality of the other figures.

- The abstract should be reduced below 300 words in length.

- Line 15 : the word « therefore » could be deleted.

- Line 38: “Abbreviations” (plural)

- Add caption “A” to figure 6

- In all micrograph figures, the length in micrometers of the bar is invisible. Please amend.

- Supplementary information, in the form of 3 tables, is limited and could easily be inserted within the main manuscript.

Author Response

Reviewer 1:

Aljobaily et al use a Methionine-Choline deficiency (MCD) dietary model in mice to induce NASH, and, using this mode study the possible beneficial effects of the non-psychotropic compound cannabigerol (CBG).

The manuscript proposes that a low dose of cannabigerol has positive effects on numerous markers of NAFLD, while not, however, ameliorating of hepatic steatosis.

Overall, this well performed investigation convincingly shows the potential benefit of CBG on the normalization of liver NASH parameters (fibrosis and inflammation), while not having a really demonstrable effect on steatosis. Also, it is noted that higher CBG doses are detrimental.

Response: The authors appreciate all the reviewer’s comments and thank the reviewer for taking the time to review the manuscript. All the comments have been addressed as suggested by the reviewer.

  1. In all the figures presenting asterisks to denote p-values, these are actually missing.

Response: The authors have addressed this concern and have added the asterisks to denote p-value.

  1. Rendering of figure 1 is quite poor. Please make sure to submit a source file of the same quality of the other figures.

Response: The authors apologize for this mistake. The quality of the figures changed when the manuscript was submitted. To address this comment, the authors sent the editor’s office the figures separately as well to ensure proper quality is presented.

  1. The abstract should be reduced below 300 words in length.

Response: the authors agree with the reviewer and have reduced the abstract length.

  1. Line 15 : the word « therefore » could be deleted.

Response: The authors apologize for this mistake. We have corrected the mistake.

  1. Line 38: “Abbreviations” (plural)

Response: The authors apologize for this mistake. We have corrected the mistake.

  1. Add caption “A” to figure 6

Response: The authors apologize for this mistake. We have added “A” to figure 6.

  1. In all micrograph figures, the length in micrometers of the bar is invisible. Please amend.

Response: The authors apologize for the inconvenience. We have increased the scale bar to make it more visible.

  1. Supplementary information, in the form of 3 tables, is limited and could easily be inserted within the main manuscript.

Response: The authors agree with the reviewer. All tables have been incorporated into the manuscript in the Methods section

Reviewer 2 Report

This preclinical study, in male (only) mice, was designed to test the hypothesis that cannabigerol (CBG) administration, after initiation of methionine/choline deficient diet (MCD), reduces structural and biochemical markers of consequent liver damage and inflammation. The model chosen is designed to mimic nonalcoholic steatohepatitis (NASH).  NASH is an important disease because its endpoint, end-stage liver disease, would lead to liver transplantation or death.

 Endpoints were hepatomegaly; fatty infiltration; histologic fibrosis, neutrophil and macrophage infiltration, expression of CB1 and CB2 receptors, and pro-inflammatory TGF-Beta 1 activity; gene expressions of mRNA for CD-36 (fat absorption), F4/80 (macrophages), and liver triglyceride content. There was no measure of the translated protein products of the respective quantitative mRNA analyses. There was also no assessment of plasma glucose, triglyceride, SGOT or SGPT (clinical enzymatic markers of hepatic injury), albumin (synthetic function), and bilirubin (clearance function).  Statistical analysis consisted of one-way ANOVA with Tukey’s post hoc testing.

Authors found that MCD diet x 5 weeks led to increased liver-to body weight ratio despite “smaller size” liver, fatty infiltration, hepatic fibrosis, hepatic infiltration by both neutrophils and macrophages, increased and more widespread hepatic expression of CB1 and CB2, and increased pro-inflammatory hepatic TGF- beta 1 mRNA expression. Three 3 weeks of intervening low-dose intraperitoneal CBG (2.46 mg/kg/day x 14 days) normalized the liver-to-body weight ratio although the pictured liver actually appears larger, from which the reader might reasonably conclude that the mouse body weight was raised relative to MCD diet alone. There was less inflammation in the low dose CBG group, evidenced histologically with or without specific staining.  Notably both neutrophil and macrophage markers were reduced in the setting of low dose CBG. Fibrosis was also improved vs MCD baseline. Both CB1 and CB2 increases were markedly attenuated by low dose CBD, and for CB2 reduced to control values. Alternatively, 14 days of high-dose CBG (24.6 mg/kg/day) paradoxically led to increased inflammatory cell infiltration both compared to other MCD mice, as well as vs no-treatment controls.  High dose CBG also worsened fibrosis in the setting of MCD diet and also when compared to control diet.  

It also failed to mimic the protective effect of low-dose CBG against increased CB1 and CB2 expression  Fatty infiltration of the liver, by several measures, was not altered by either CBG treatment protocol. Neither dose of CBG had any effect on control liver expression of CB1 or CB2. In the discussion, the authors admit that the phenotype of NASH, which includes metabolic syndrome (and typically obesity) was not fully paralleled by the MCD-diet male mouse model. The authors conclude that low-dose, but not high dose, CBG treatment improves several markers of NASH.     

Critique: The title is misleading in two respects. First, it states “symptoms” of NASH, which by definition is clinical complaints reported or experienced by the patient rather than “evidence” or “signs” which might include pathological findings. And second, it fails to identify that the animals used are male (only) mice.  These both must be corrected. Authors also fail in manuscript to discriminate between NASH and its precedent nonalcoholic fatty liver disease (NAFLD). Finally, since purpose of the submission is to show that progression of the relevant NASH human disease is improved by CBG administration, parallel systemic indices that should be included are plasma concentrations of SGOT and SGPT (clinical enzymatic markers of hepatic injury), albumin (synthetic function), and bilirubin (clearance function). Also missing is the plasma glucose and insulin concentrations, which is important because cirrhosis leads to profound insulin resistance and consequent hyperglycemia which ultimately become part of the clinical NASH syndrome. Further the authors did not measure circulating triglyceride concentration, as is done in NASH patients.  Hepatic triglyceride assay should be presented in the methods, not just the results. The correlation between plasma triglyceride concentration used clinically to evaluate NASH in patients, which was no performed by the authors, and the hepatic content of triglyceride presented in Fig 2 is not clear.  Figure 4 legend implies that alpha-SMA gene expression (mRNA, a surrogate for collagen presence) is portrayed but the actual related graphic has evidently been banished into “Supplementary Figure 3” which is not visible to readers. Authors must acknowledge that failure to include female mice is weakness of the study that could have serious ramifications. Organization of the results presentation is flawed in that the sequence of data is odd, with “overall” results confusingly interposed with specific datasets. There should be regular logical presentation of the effect of the MCD diet without intervention.  No data are presented for mouse body size or food intake, which makes Fig 1 “liver to body weight ratio” impossible to interpret. Do they really have hepatomegaly or is the MCD-treated mouse failure to thrive a larger impact than the actually smaller than control liver size? Finally, the mRNA data are not accompanied by actual protein data to indicate that the associated translational products are also changed in the same direction. There is some question of whether the MCD-diet actually models the clinical syndrome of NASH, as the animals are not obese and do not show evidence of metabolic syndrome (hypertension, insulin resistance/hyperglycemia, and obesity), factors which were not critically assessed in the present study.  

Author Response

Reviewer 2:

This preclinical study, in male (only) mice, was designed to test the hypothesis that cannabigerol (CBG) administration, after initiation of methionine/choline deficient diet (MCD), reduces structural and biochemical markers of consequent liver damage and inflammation. The model chosen is designed to mimic nonalcoholic steatohepatitis (NASH).  NASH is an important disease because its endpoint, end-stage liver disease, would lead to liver transplantation or death.

 Endpoints were hepatomegaly; fatty infiltration; histologic fibrosis, neutrophil and macrophage infiltration, expression of CB1 and CB2 receptors, and pro-inflammatory TGF-Beta 1 activity; gene expressions of mRNA for CD-36 (fat absorption), F4/80 (macrophages), and liver triglyceride content. There was no measure of the translated protein products of the respective quantitative mRNA analyses. There was also no assessment of plasma glucose, triglyceride, SGOT or SGPT (clinical enzymatic markers of hepatic injury), albumin (synthetic function), and bilirubin (clearance function).  Statistical analysis consisted of one-way ANOVA with Tukey’s post hoc testing.

Authors found that MCD diet x 5 weeks led to increased liver-to body weight ratio despite “smaller size” liver, fatty infiltration, hepatic fibrosis, hepatic infiltration by both neutrophils and macrophages, increased and more widespread hepatic expression of CB1 and CB2, and increased pro-inflammatory hepatic TGF- beta 1 mRNA expression. Three 3 weeks of intervening low-dose intraperitoneal CBG (2.46 mg/kg/day x 14 days) normalized the liver-to-body weight ratio although the pictured liver actually appears larger, from which the reader might reasonably conclude that the mouse body weight was raised relative to MCD diet alone. There was less inflammation in the low dose CBG group, evidenced histologically with or without specific staining.  Notably both neutrophil and macrophage markers were reduced in the setting of low dose CBG. Fibrosis was also improved vs MCD baseline. Both CB1 and CB2 increases were markedly attenuated by low dose CBD, and for CB2 reduced to control values. Alternatively, 14 days of high-dose CBG (24.6 mg/kg/day) paradoxically led to increased inflammatory cell infiltration both compared to other MCD mice, as well as vs no-treatment controls.  High dose CBG also worsened fibrosis in the setting of MCD diet and also when compared to control diet. 

It also failed to mimic the protective effect of low-dose CBG against increased CB1 and CB2 expression  Fatty infiltration of the liver, by several measures, was not altered by either CBG treatment protocol. Neither dose of CBG had any effect on control liver expression of CB1 or CB2. In the discussion, the authors admit that the phenotype of NASH, which includes metabolic syndrome (and typically obesity) was not fully paralleled by the MCD-diet male mouse model. The authors conclude that low-dose, but not high dose, CBG treatment improves several markers of NASH.    

Response: The authors appreciate all the reviewer’s comments and thank the reviewer for taking the time to review the manuscript. All the comments have been addressed as suggested by the reviewer.

  1. The title is misleading in two respects. First, it states “symptoms” of NASH, which by definition is clinical complaints reported or experienced by the patient rather than “evidence” or “signs” which might include pathological findings.

Response: The authors appreciate the suggestion. The title has been changed from [Low dose administration of Cannabigerol attenuates methionine/choline deficient diet-induced NASH symptoms via modulation of cannabinoid receptor] to [Low dose administration of Cannabigerol attenuates inflammation and fibrosis associated with methionine/choline deficient diet-induced NASH symptoms via modulation of cannabinoid receptor].

  1. And second, it fails to identify that the animals used are male (only) mice.  These both must be corrected. Authors also fail in manuscript to discriminate between NASH and its precedent nonalcoholic fatty liver disease (NAFLD).

Response: The authors apologize for not justifying the reason behind using only male mice. The justification has been added to the manuscript in the discussion section. [In this study, we only analyzed the male mice so far. The therapeutic effect of CBG in female needs further investigation. In the exploratory experiments, we have found gender differential responses to the CBG in terms of alleviation of inflammation and fibrosis. However, further experiments needs to be done to confirm the results.]

  1. Finally, since purpose of the submission is to show that progression of the relevant NASH human disease is improved by CBG administration, parallel systemic indices that should be included are plasma concentrations of SGOT and SGPT (clinical enzymatic markers of hepatic injury), albumin (synthetic function), and bilirubin (clearance function). Also missing is the plasma glucose and insulin concentrations, which is important because cirrhosis leads to profound insulin resistance and consequent hyperglycemia which ultimately become part of the clinical NASH syndrome.

Response: The authors appreciate the concern raised by the reviewer. The authors would like to clarify that the MCD diet does not induce insulin resistance. In addition, the authors did perform a full liver panel using the serum of the mice. However due to high hemolysis background, the results were not even close to the reference values due to the noise from the heolysis. Even though we are not able to get informative results from liver panel. We showed improved H&E appearance and liver to body weight ratio and other direct proof from liver tissue. As indicated in discussion, we also pointed out that the MCD diet could not fully recapitulate the metabolic syndrome in mice. In order to better reflecting our results, we have modified the title to emphasize the effect on reducing inflammation and fibrosis, which can see in human NASH.

  1. Further the authors did not measure circulating triglyceride concentration, as is done in NASH patients.  Hepatic triglyceride assay should be presented in the methods, not just the results. The correlation between plasma triglyceride concentration used clinically to evaluate NASH in patients, which was no performed by the authors, and the hepatic content of triglyceride presented in Fig 2 is not clear.  

Response: Based on the previous study1, the circulating triglyceride will not show significant difference between control and MCD diet as the diet will prevent the fatty acids being transported outside of the liver, which is also the reason of significant body weight loss from these mice. Therefore, we did not perform the circulating triglyceride measurement since the diet will not cause the elevated levels of triglyceride in serum. This is an aggressive diet to cause acute hepatic inflammation and fibrosis due to hepatotoxicity. In terms of hepatic triglyceride assay, we apologize that not putting the methods for this part. We have added a new session “2.7” for the triglyceride assay.

  1. Figure 4 legend implies that alpha-SMA gene expression (mRNA, a surrogate for collagen presence) is portrayed but the actual related graphic has evidently been banished into “Supplementary Figure 3” which is not visible to readers. Authors must acknowledge that failure to include female mice is weakness of the study that could have serious ramifications. Organization of the results presentation is flawed in that the sequence of data is odd, with “overall” results confusingly interposed with specific datasets. There should be regular logical presentation of the effect of the MCD diet without intervention.

Response: The authors apologize for this mistake. The mRNA level graph has been incorporated into the manuscript in figure 4. We also added the weakness of the results in terms of MCD diet and gender bias to the discussion. In terms of the logical flow of data, we intended to show the liver morphology and change of liver and body weight first, and then follow the disease potential symptoms, to evaluate the steatosis, inflammation, and fibrotic changes. And then we want to dig deeper to investigate the potential interaction of CBG with CB receptors and the modulation of these receptors upon the treatment of CBG. And how immune cell subpopulation get enrolled in this process.

  1. No data are presented for mouse body size or food intake, which makes Fig 1 “liver to body weight ratio” impossible to interpret. Do they really have hepatomegaly or is the MCD-treated mouse failure to thrive a larger impact than the actually smaller than control liver size?

Response: The authors apologize for this mistake. The body, liver weight and food consumption has been provided as supplemental figure 1. Both liver and body weight decreased upon the treatment of MCD. However, the ratio of liver weight to body weight ratio recovered with low dose of CBG. To respectively answer reviewer’s question, the liver weight did not increase upon the treatment of MCD, smaller in size. The low dose of CBG showed trend of decreased liver weight when compared to MCD which makes the MCD showed higher liver to body weight ratio.

  1. Finally, the mRNA data are not accompanied by actual protein data to indicate that the associated translational products are also changed in the same direction.

Response: The authors appreciate the reviewer’s comment. Protein expression is presented in the immunofluorescence staining performed on liver sections. And now the mRNA data is in the same figure as the protein data.

  1. There is some question of whether the MCD-diet actually models the clinical syndrome of NASH, as the animals are not obese and do not show evidence of metabolic syndrome (hypertension, insulin resistance/hyperglycemia, and obesity), factors which were not critically assessed in the present study.  

Response: The authors appreciate the comment suggested by the reviewer. The authors are aware of this as a weakness of the MCD diet. The authors clarified this concern in the discussion section. [It is important to acknowledge that the MCD diet does not fully replicate the human NASH symptoms. The mouse MCD diet fails to show metabolic syndrome, insulin resistance and body weight gain, it replicates other NASH symptoms such as hepatic steatosis, inflammation, ER stress, cellular apoptosis and fibrosis over a short period. It is critical to keep in mind that the MCD diet does not induce glucose intolerance, nor it replicates the human metabolic syndrome associated with NASH]

  1. Rinella ME, Green RM. The methionine-choline deficient dietary model of steatohepatitis does not exhibit insulin resistance. J Hepatol. 2004;40(1):47-51.

Reviewer 3 Report

Comments to the Author

The manuscript summarizes the therapeutic potential of CBG in reducing hepatic steatosis, fibrosis, and inflammation and demonstrates the protective effect of CBG in MCD diet-induced NASH in mice. This study is meaningful and helps to make clear the pathological process of NASH. However, there exist several format and grammatical issues. Generally, the manuscript is well structured and the contents are rich.  The specific issues are mentioned below.

Specific comment:

1.  Line 63-74, in this study, Cannabigerol (CBG) was used to attenuate NASH symptoms, but the authors overstated the progress of CBD, making the introduction redundant.

2.  Line 90, use the abbreviation when it appears a second time. Please check and correct the full text.

3.  Line 101, “primers were synthesized by”

4.  Line 144, change [ten minutes] to [10 minutes]. Please check throughout the text.

5.  Line 156, delete the comma.

6.  Line 172, the grammar of “The Shapiro-Wilk test was performed to test for normality before parametric statistical tests were used” is not correct.

7.  Line 230, Figure 2B, 2C, 2D and Figure 3B, 3D, 3E lack the significant analysis. Please check throughout the figures.

8.  Line344, delete the point.

9.  Line 349, the grammar of “it is becoming increasingly important to investigate alternative treatments to slow the progression of the disease” is not correct.

10.  Line 394-395, the high dose (24.6 mg/kg) is 10 times higher than the low dose (2.46 mg/kg). So, the bad effect of CBG treatment in high doses may be due to its excessive concentration. Why did you choose these two concentrations, please explain.

Author Response

Reviewer 3:

The manuscript summarizes the therapeutic potential of CBG in reducing hepatic steatosis, fibrosis, and inflammation and demonstrates the protective effect of CBG in MCD diet-induced NASH in mice. This study is meaningful and helps to make clear the pathological process of NASH. However, there exist several format and grammatical issues. Generally, the manuscript is well structured and the contents are rich.  The specific issues are mentioned below.

Response: The authors appreciate all the reviewer’s comments and thank the reviewer for taking the time to review the manuscript. All the comments have been addressed as suggested by the reviewer.

  1. Line 63-74, in this study, Cannabigerol (CBG) was used to attenuate NASH symptoms, but the authors overstated the progress of CBD, making the introduction redundant.

Response: The authors agree with the reviewer. The details of CBD treatment has been removed to just emphasize the related research about CBG since this study did not focus on CBD.

  1. Line 90, use the abbreviation when it appears a second time. Please check and correct the full text.

Response: The authors apologize for this mistake. All the names have been changed to abbreviations when used for the second time.

  1. Line 101, “primers were synthesized by” done

Response: The authors apologize for this mistake. The sentence was changed from […while all reagents and primers were purchased from Thermo Fisher Scientific…] to [while all reagents and primers were synthesized by Thermo Fisher Scientific…].

  1. Line 144, change [ten minutes] to [10 minutes]. Please check throughout the text.

Response: The authors apologize for this mistake. The word was changed from [ten minutes] to [10 minutes].

  1. Line 156, delete the comma.

Response: The authors have deleted the comma.

  1. Line 172, the grammar of “The Shapiro-Wilk test was performed to test for normality before parametric statistical tests were used” is not correct. Done- deleted the whole sentence

Response: The authors apologize for this mistake. This sentence was deleted.

  1. Line 230, Figure 2B, 2C, 2D and Figure 3B, 3D, 3E lack the significant analysis. Please check throughout the figures.

Response: the authors apologize for this mistake. The significant asterisks were added to all the figures.

  1. Line344, delete the point.

Response: The authors have deleted the point.

  1. Line 349, the grammar of “it is becoming increasingly important to investigate alternative treatments to slow the progression of the disease” is not correct.

Response: the authors apologize for this mistake. The sentence was changed from […it is becoming increasingly important to investigate alternative treatments to slow the progression of the disease…] to [While lifestyle changes and invasive surgeries are the only approved clinical approaches to slow the progression of NASH, it is critical to investigate alternative treatments and strategies to slow the progression of the disease]

  1. Line 394-395, the high dose (24.6 mg/kg) is 10 times higher than the low dose (2.46 mg/kg). So, the bad effect of CBG treatment in high doses may be due to its excessive concentration. Why did you choose these two concentrations, please explain.

Response: the authors agree with the reviewer about the need to justify the concentrations used in the study. The high dose concentration was based on one-tenth of the human maximum allowed daily consumption of CBD and then converted into mice based on previously published study.1 The low dose concentration were chosen based on previously published study.2

  1. Wojcikowski K, Gobe G. Animal studies on medicinal herbs: predictability, dose conversion and potential value. Phytother Res. 2014;28(1):22-27.

2.Zagzoog, A., Mohamed, K.A., Kim, H.J. et al. In vitro and in vivo pharmacological activity of minor cannabinoids isolated from Cannabis sativa. Sci Rep 10, 20405 (2020).

Round 2

Reviewer 2 Report

This revised preclinical study, in male (only) mice, was designed to test the hypothesis that cannabigerol (CBG) administration, after initiation of methionine/choline deficient diet (MCD), reduces structural and biochemical markers of consequent liver damage and inflammation. The model chosen is designed to mimic nonalcoholic steatohepatitis (NASH). NASH is an important disease because its endpoint, end-stage liver disease, would lead to liver transplantation or death.

Endpoints were liver size-to body weight ratio; fatty infiltration; histologic fibrosis, neutrophil and macrophage infiltration, expression of CB1 and CB2 receptors, and pro-inflammatory TGF-Beta 1 activity; gene expressions of mRNA for CD-36 (fat absorption), F4/80 (macrophages), and liver triglyceride content. There was no measure of the translated protein products of the respective quantitative mRNA analyses. There was also no assessment of plasma glucose, triglyceride, SGOT or SGPT (clinical enzymatic markers of hepatic injury), albumin (synthetic function), and bilirubin (clearance function). Statistical analysis consisted of one-way ANOVA with Tukey’s post hoc testing.

Authors found that MCD diet x 5 weeks led to increased liver-to body weight ratio despite “smaller size” liver, fatty infiltration, hepatic fibrosis, hepatic infiltration by both neutrophils and macrophages, increased and more widespread hepatic expression of CB1 and CB2, and increased pro-inflammatory hepatic TGF- beta 1 mRNA expression. Three 3 weeks of intervening low-dose intraperitoneal CBG (2.46 mg/kg/day x 14 days) normalized the liver-to-body weight ratio although the pictured liver actually appears larger, from which the reader might reasonably conclude that the mouse body weight was raised relative to MCD diet alone. There was less inflammation in the low-dose CBG group, evidenced histologically with or without specific staining. Notably both neutrophil and macrophage markers were reduced in the setting of low dose CBG. Fibrosis was also improved vs MCD baseline. Both CB1 and CB2 increases were markedly attenuated by low dose CBD, and for CB2 reduced to control values. Alternatively, 14 days of high-dose CBG (24.6 mg/kg/day) paradoxically led to increased inflammatory cell infiltration both compared to other MCD mice, as well as vs no-treatment controls. High dose CBG also worsened fibrosis in the setting of MCD diet and also when compared to control diet. High-dose CBG also failed to mimic the protective effect of low-dose CBG against increased CB1 and CB2 expression Fatty infiltration of the liver, by several measures, was not altered by either CBG treatment protocol. Neither dose of CBG had any effect on control liver expression of CB1 or CB2. In the discussion, the authors admit that the phenotype of NASH, which includes metabolic syndrome (and typically obesity) was not fully paralleled by the MCD-diet male mouse model. The authors conclude that low-dose, but not high dose, CBG treatment improves several markers of NASH. 

Critique:  The title is now corrected to this reviewer’s satisfaction.  

Authors have improved their discrimination between NASH and its precedent nonalcoholic fatty liver disease (NAFLD). They have improved the flow of information.

Authors have corrected Figure 4 graphic and its associated legend.

Remaining issues:

Authors explained in their cover letter why they failed to report plasma markers of the systemic manifestations of NASH in their model human as hemolysis during blood sample collection. This is important information, and would have been useful to detect systemic disease markers which might have been is improved by CBG administration. As noted in my initial review, systemic indices that should be included are plasma concentrations of SGOT and SGPT (clinical enzymatic markers of hepatic injury), albumin (synthetic function), and bilirubin (clearance function).  Authors should include in the manuscript text that this is a weakness of the paper, but imply that not one that they had overlooked.  They should also make clear in the discussion that plasma glucose and insulin concentrations, while important because cirrhosis leads to profound insulin resistance and consequent hyperglycemia which ultimately become part of the clinical NASH syndrome, were less important in their tested model because these features are nor produced by the methionine/choline deficient diet. Expand on the limitations of male-only testing in the discussion.

The phrase liver-to body weight ratio usually implies changes in liver size, rather than body wasting due to adipose loss.  Figure 1 still must be reconstructed to incorporate body weight, rather than burying body weight in Supplementary Figure 1B, to improve readability.  Food intake could be also presented in either the text or as part of Fig 1 (not Supplemental Fig 1A), for clarity.      

Author Response

Comment 1: Authors explained in their cover letter why they failed to report plasma markers of the systemic manifestations of NASH in their model human as hemolysis during blood sample collection. This is important information, and would have been useful to detect systemic disease markers which might have been is improved by CBG administration. As noted in my initial review, systemic indices that should be included are plasma concentrations of SGOT and SGPT (clinical enzymatic markers of hepatic injury), albumin (synthetic function), and bilirubin (clearance function).  Authors should include in the manuscript text that this is a weakness of the paper, but imply that not one that they had overlooked.  

Reply: We appreciate reviewer’s comments and suggestions. We have added this as a weakness in the discussion. [It’s important to evaluate the systemic disease markers, such as bilirubin, Glutamic-oxalacetic transaminase and glutamic-pyruvic transaminase, for the therapeutic potential of CBG to improve liver function. Unfortunately, the collected data was not informative due to severe hemolysis (data not shown). ]

They should also make clear in the discussion that plasma glucose and insulin concentrations, while important because cirrhosis leads to profound insulin resistance and consequent hyperglycemia which ultimately become part of the clinical NASH syndrome, were less important in their tested model because these features are nor produced by the methionine/choline deficient diet. Expand on the limitations of male-only testing in the discussion.

Reply: We appreciate the comments. The glucose and insulin resistance is very important symptoms showing in human NAFLD and NASH. We have discussed this importance on line 350-362. We also point out the limitations of male only testing in line 358-360.

The phrase liver-to body weight ratio usually implies changes in liver size, rather than body wasting due to adipose loss.  Figure 1 still must be reconstructed to incorporate body weight, rather than burying body weight in Supplementary Figure 1B, to improve readability.  Food intake could be also presented in either the text or as part of Fig 1 (not Supplemental Fig 1A), for clarity.     

Reply: The figure 1 is updated according to reviewer’s suggestion.

Reviewer 3 Report

This reviewer accepted the revised version  

Author Response

We appreciate reviewer's comments